# Association between mobility, non-pharmaceutical interventions, and COVID-19 transmission in Ghana: A modelling study using mobile phone data

**Hamish Gibbs**[1]*, **Yang Liu**[1], **Sam Abbott**[1], **Isaac Baffoe-Nyarko**[2], **Dennis O. Laryea**[2], **Ernest Akyereko**[2], **Patrick Kuma-Aboagye**[2], **Ivy Asantewaa Asante**[3], **Oriol Mitjà**[4], **LSHTM CMMID COVID-19 Working Group**[¶], **William Ampofo**[3], **Franklin Asiedu-Bekoe**[2], **Michael Marks**[5,6,7], **Rosalind M. Eggo**[1]

1 Department of Infectious Disease Epidemiology, London School of Hygiene & Tropical Medicine, London, United Kingdom, 2 Ghana Health Service, Ministry of Health, Accra, Ghana, 3 Noguchi Memorial Institute for Medical Research, Accra, Ghana, 4 Fight AIDS and Infectious Diseases Foundation, Hospital Universitari Germans Trias i Pujol, Badalona, Spain, 5 Department of Clinical Research, London School of Hygiene & Tropical Medicine, London, United Kingdom, 6 Hospital for Tropical Diseases, University College London Hospital, London, United Kingdom, 7 Division of Infection and Immunity, University College London, London, United Kingdom

¶ Membership of the LSHTM CMMID COVID-19 Working Group is listed in the Acknowledgments.
* Hamish.Gibbs.21@ucl.ac.uk

**Data Availability Statement:** Data used in this study included individual Line List data shared with authors by the Ghana Health Service. Use of this

## Abstract

Governments around the world have implemented non-pharmaceutical interventions to limit the transmission of COVID-19. Here we assess if increasing NPI stringency was associated with a reduction in COVID-19 cases in Ghana. While lockdowns and physical distancing have proven effective for reducing COVID-19 transmission, there is still limited understanding of how NPI measures are reflected in indicators of human mobility. Further, there is a lack of understanding about how findings from high-income settings correspond to low and middle-income contexts. In this study, we assess the relationship between indicators of human mobility, NPIs, and estimates of $R_t$, a real-time measure of the intensity of COVID-19 transmission. We construct a multilevel generalised linear mixed model, combining local disease surveillance data from subnational districts of Ghana with the timing of NPIs and indicators of human mobility from Google and Vodafone Ghana. We observe a relationship between reductions in human mobility and decreases in $R_t$ during the early stages of the COVID-19 epidemic in Ghana. We find that the strength of this relationship varies through time, decreasing after the most stringent period of interventions in the early epidemic. Our findings demonstrate how the association of NPI and mobility indicators with COVID-19 transmission may vary through time. Further, we demonstrate the utility of combining local disease surveillance data with large scale human mobility data to augment existing surveillance capacity to monitor the impact of NPI policies.

data was approved by the LSHTM Research Committee (Ref: 22477) and the Noguchi Memorial Institute of Medical Research (Ref: 048/20-21). Vodafone mobility data used in this study is proprietary data shared by Vodafone Ghana in partnership with the Flowminder Foundation and Ghana Statistical Service. This mobility data is available to researchers by application. Any other researchers who wish to use this data can apply to the FlowMinder foundation: https://www.flowminder.org/. Google mobility data used in this study is available in the public domain. Downloads of this data can be found in the references. Code used in this study is available from: https://github.com/hamishgibbs/ghana_rt_npi_mobility.

**Funding:** The following funding sources are acknowledged as providing funding for the named authors. This research was partly funded by the Bill & Melinda Gates Foundation (INV-003174: YL). EDCTP2 (RIA2020EF-2983-CSIGN: HPG, OM, RME, MM). This project is part of the EDCTP2 programme supported by the European Union (RIA2020EF-2983-CSIGN: HPG, OM, RME, MM). This project has received funding from the European Union's Horizon 2020 research and innovation programme - project EpiPose (101003688: YL). HDR UK (MR/S003975/1: RME). This research was partly funded by the National Institute for Health Research (NIHR) using UK aid from the UK Government to support global health research. The views expressed in this publication are those of the author(s) and not necessarily those of the NIHR or the UK Department of Health and Social Care (16/137/109: YL; NIHR200908: RME). UK DHSC/UK Aid/NIHR (PR-OD-1017-20001: HPG). UK MRC (MC_PC_19065: RME, YL). Wellcome Trust (210758/Z/18/Z: SA). The funders had no role in study design, data collection and analysis, decision to publish, or preparation of the manuscript.

**Competing interests:** The authors declare no competing interests.

## Introduction

Nations around the world introduced a range of non-pharmaceutical interventions (NPIs) to limit the spread of COVID-19 in the early phases of the epidemic [1]. In Ghana, NPIs have been diverse, and have included the use of personal protective measures, environmental measures, physical distancing, restricting movement, and limiting the gathering of people. NPIs have been implemented at different times in relation to the progression of local and national disease outbreaks, with some put in place before transmission was established, and others reactive to rises in cases. NPI measures have also overlapped one another in the timing of their application [1, 2]. Previous research has attempted to quantify the relative effectiveness on COVID-19 transmission of different NPIs [3–6], but modelling the impact of different intervention strategies includes uncertainty about how different strategies are implemented in practice. Additionally, statistical approaches for estimating the impact of individual interventions can be confounded by the overlapping nature of NPI policies and the different mechanisms that interventions use to reduce disease transmission. There remain significant open questions about methods for reliably isolating and quantifying the individual effect of each intervention [7].

One approach used by researchers and policymakers to measure the impact of NPIs during the COVID-19 pandemic has been to observe changes in measurements of human behaviour under individual interventions or under a combination of interventions [8–12]. Perhaps the most common way to quantify varying patterns of human behaviour is the use of human mobility datasets, which measure the locations of individuals using GPS or Call Detail Records (CDRs) [13, 14]. These mobility datasets have been made available by a variety of network service and mobile application providers [15–17]. Mobility data has been used widely during the COVID-19 pandemic to predict the introduction of COVID-19 cases, and to monitor and estimate adherence to NPIs including travel restrictions [9, 18–20], but questions remain about how patterns of mobility and NPI stringency relate to transmission in LMIC settings.

Previous research has been conducted in Africa on the implications of mobility patterns for transmission of infections other than COVID-19 [21, 22] and during the COVID-19 epidemic, analysis of movement patterns in Ghana has been conducted to inform policy makers about the volume of reductions coinciding with lockdown interventions in Accra and Kumasi [23]. These indicators may be used as a proxy for social contact [13] and therefore, for potential COVID-19 transmission, although the "link" between movement and disease transmission may decrease due to greater adherence to social distancing or personal protective equipment guidelines [24]. There remain questions about how mobility indicators can be used to estimate COVID-19 transmission and how these indicators reflect behavioural responses to NPI measures, particularly in an LMIC context. Here we estimate the relationship between the transmission of COVID-19 in Ghana and the changing NPIs introduced to mitigate the epidemic. We also estimate the effect of NPIs on transmission using human mobility data, which is a measurable proxy for responses to NPIs.

## Methods

### Study setting

The first cases of COVID-19 in Ghana were reported on 12th March 2020 [25]. These cases were detected in Accra, the capital city of Ghana and were imported via international travel [25]. Following the announcement of the first COVID-19 cases, the Ghanaian government announced the suspension of international travel and the closure of land borders to reduce the risk of further introduction [26]. Domestic case numbers grew in March and April 2020,

leading to the closure of universities and high schools and the announcement of a partial lock-down in the Ashanti and Greater Accra regions, the two most populous regions of Ghana [26]. This lockdown introduced a stay at home order except for essential travel including shopping, healthcare, and use of public toilets. Almost all COVID-19 NPI restrictions were lifted by July, although restrictions on international travel and mandated use of facemasks remained in place until September 2020.

## COVID-19 surveillance data

We used line list surveillance data recording lab-confirmed COVID-19 cases in a sample of 27 districts collected by the Ghana Health Service on PCR confirmed COVID-19 patients at the district level (administrative level 2, 261 total districts) in 11 of the 16 regions of Ghana between March and September 2020 These data were collected in 11 of the 16 regions in Ghana (administrative level 1: excluding "Ahafo", "Bono", "Upper West", "Volta", and "Western North" districts due to limited detail of data in these districts). Patient-level records were referenced to a standard spatial reference provided by the Ghana Statistical Service using patients' reported district of residence. Using the date of case confirmation and the district of residence, we aggregated individual records into daily case counts of confirmed COVID-19 cases per district (Figure A in S1 Text). Through visual inspection, we replaced three outliers in reporting in two districts with a linear interpolation between the preceding and following records. For these outliers, the number of cases reported in a district clearly exceeded the overall trend of case reporting (reported cases greater than 5x higher than all previous reports) (Table A in S1 Text). We assumed that these records reflected "late reporting" with samples collected on multiple days reported on the same date. This assumption potentially underestimates the total number of COVID-19 cases in these two districts, but it is not possible to approximate when the cases reported in these intervals may have been originally tested.

## Defining stringency indices for NPIs

Data on the dates of NPIs implemented in Ghana were provided by Ghana Health Service detailing the starting dates of public health interventions. We used this to define the start dates of interventions and augmented it with available news sources and government press releases to create a dataset of the start and end dates of nine intervention measures. Using these intervention data, we constructed a stringency index to measure the stringency of COVID-19 interventions through time, defined daily as the number of active interventions divided by the total number of interventions. This stringency index assigns a uniform level of stringency to each intervention measure and records the length of time that the measure was implemented. We also used the OxCGRT stringency index, calculated from a global database of NPIs, which is used to construct a stringency index based on a taxonomy of government interventions [1] (Figure B in S1 Text). We used the most recent version of the stringency index (as of June 2021), rather than the OxCGRT "legacy stringency index." OxCGRT data also records nine interventions resulting in a change in the stringency index in the study period. We extracted the date of maximal intervention from both indices to compare both stringency indicators, defined as the first date with the highest stringency for each index. Although both sources of intervention data reported interventions at a national level, intervention measures were introduced at different spatial scales in Ghana. School closures (including different educational tracks) and mask mandates, for example, were implemented nationally, while partial lockdown measures were introduced only in Ashanti and Greater Accra regions.

## $R_t$ estimation

Rt is a time-varying parameter describing the average number of infections derived from a single infection and indicates whether an epidemic is growing (Rt > 1) or decreasing (Rt < 1).

COVID-19 testing was not equally available in all parts of Ghana during the COVID-19 epidemic and line list data includes many districts without complete time series in the early epidemic. Any testing arriving into a community after the introduction of COVID-19 may be biased by reporting both existing and newly acquired infections. To ensure the quality of data during the early stages of the COVID-19 outbreak in Ghana, we limited $R_t$ estimation to 27 districts with reported cases before March 30th, 2020 (the beginning of the partial lockdown in Ashanti and Greater Accra), and at least 100 reported cases during the entire study period (12th March to 1st September 2020). We chose this threshold to select districts with surveillance resources capable of detecting cases in the early epidemic, due to uncertainty about whether case reporting followed the path of the epidemic or the availability of testing in districts with irregular case reporting.

$R_t$ estimates were calculated at the district level using the EpiNow2 R package (1.3.2) using MCMC, as implemented in Stan [27, 28], based on weekly reported cases [29]. Expected daily cases were estimated using the renewal equation to weight prior expected cases multiplied by the estimated $R_t$. Variation in $R_t$ over time was modelled using a mean intercept and an approximate Gaussian process with a 3/2 matern kernel on the log scale [30, 31]. Unlike in Abbott et al. [30] we modelled $R_t$ explicitly with the gaussian process and not as a first order difference. This has little impact on retrospective $R_t$ estimates and substantially reduces the computational overhead. We used a generation time modelled as a gamma distribution with mean: 3.6 (standard deviation of mean: 0.7), standard deviation 3.1 (standard deviation of standard deviation: 0.76) and maximum: 15 [5, 32]. We assumed a negative binomial observation model for reported cases with a day of the week effect modelled as a simplex allowing us to model weekly reported cases without manual specification. $R_t$ estimates did not include an estimate of reporting delays as lags were estimated in subsequent analyses. Therefore, estimates of Rt on a given date vary as a result of the reported cases on that date. Inference was performed across 4 chains for 2000 samples with a burn-in period of 250 samples. Convergence was diagnosed using the R hat diagnostic [28].

## Defining a mobility indicator from Vodafone data

We used Vodafone Ghana Call Detail Records (CDRs) aggregated by the Flowminder Foundation prior to data sharing [33]. CDRs record mobile phone connections to the cellular towers routing a call or SMS. CDRs are used by mobile network providers for billing purposes. CDRs provide the location of the mobile phone and SIM card based on the location of the cellular tower routing the signal, most often the nearest one. The precision of estimated mobile phone locations depends on the density of cellular towers in an area and signal coverage. It can reach up to 3 km in average coverage conditions and up to 8 km in good coverage conditions. Individual mobility can be estimated by recording a series of mobile phone connections to cellular towers over time. We used CDR data aggregated into an origin-destination matrix, based on the locations recorded within 24 hours for individual mobile phones. The data were censored to remove daily counts of 10 or fewer subscribers recorded for an origin-destination pair, in order to reduce the risk of statistical disclosure of personally identifiable information.

We generated a normalised index of movement outside of individual districts relative to baseline values from the origin-destination matrix. We used two metrics to calculate this normalised measure of mobility: (1) trips between districts: the daily number of subscribers travelling between pairs of districts, and (2) total subscribers per district: the total number of unique

mobile phone subscribers recorded in a district on each day. Because of inconsistencies between the spatial references used for the mobility data and the case data, we aggregated mobility indicators for Accra, Tema, and Kumasi Metropolitan Areas by removing trips between aggregated districts and calculating the sum of subscribers for these districts. The mobility data also contained 10 missing dates (4.1%) and we performed linear interpolation for each district for both metrics (trips and subscriber counts) on these dates (Figure C in S1 Text).

To construct the normalised movement index we summed the total number of outgoing trips for individual districts on each day. These values were then normalised by the total number of daily subscribers in individual districts to remove bias introduced solely because of varying numbers of subscribers. For each district $i$ and each day $t$, the normalised number of outgoing trips was defined as:

$$trips\_out\_norm_{i,t} = \frac{trips\_out_{i,t}}{total\_subscribers_{i,t}}$$

The movement index measures the change in outbound trips from individual districts relative to baseline values using September to December 2019 as the baseline period. We chose this baseline as it includes the earliest period for which mobility data was available, but this baseline may not account for seasonal variations in movement patterns during a year.

Baseline values were then calculated per week day during the baseline period as the median of outgoing trips (normalised by the number of subscribers, as above) for each district $i$ and each day of the week, $j$:

$$baseline_{i,j} = median(trips\_out\_norm_{i,j})$$

Using the baseline values, we calculated the deviation from baseline in the study period as a percentage for each district $i$ on each day $t$ given the day of week $j$ of $t$:

$$percent\_change_{i,t} = \frac{(observed_{i,t} - baseline_{i,j})}{baseline_{i,j}} * 100$$

This resulted in a normalised mobility indicator (Figure D in S1 Text).

## Google mobility indicator

We used mobility data from Google as a second measure of human movement [16]. This data records the GPS location of individuals actively using Google services who have chosen to share their location data with Google. The data is provided as a measure of changes in activity relative to a baseline in different settings (Residential, Grocery & Pharmacy, Retail & Recreation, Transit Stations, Workplaces). The dataset documentation recommends consideration of the specifics of mobility in different settings. We chose to use only the mobility indicator from the "Residential" setting because we considered this setting to be the most clearly defined setting in the context of Ghana, and because of the relatively lower variance of this indicator (Figure E in S1 Text). We calculated the inverse of the percent change in residential mobility. This percentage is relative to a baseline period between 3rd January and 6th February, 2020 which is defined by Google prior to data sharing.

Google mobility data is not referenced to known administrative areas but rather to custom boundary polygons, which do not closely align with administrative districts in Ghana. To combine this mobility data with the other data sources used in this study, we manually digitized (traced) these features to create a spatial representation of the coverage area of each metric. To

align Google mobility data with our spatial reference, we assigned Google mobility data to those districts with greater than 50% overlap with the administrative areas defined by Google for Accra and Kumasi. This restricted the coverage of Google mobility data to central districts in the Accra and Kumasi Metropolitan Areas (Figure F in S1 Text).

## Statistical analysis

We assessed the association between NPIs, mobility and median $R_t$ while adjusting for public holidays using a two-level multilevel generalised linear mixed model (using a Gaussian observation model), with random intercepts for individual districts to account for local variation. We used data from 12th March 2020 to 1st September 2020, during the period of interventions in Ghana, and before the detection of the Alpha or Beta variants [34].

Level 1 of the multi-level model included the relationship between mobility indicators and the inverse NPI stringency with district-level random effects, $x$ to account for correlation between mobility and NPI stringency:

Level 1: Mobility$_a$ ~ x$_a$ + NPI stringency

Where $a$ indicates individual districts.

Level 2 modelled the relationship between $R_t$, NPI stringency, residuals from Level 1, and district-level random effects:

Level 2: Rt$_a$ ~ x$_a$ + NPI Stringency$_{, i\, -j}$ + residuals(Level 1$_{a,\, i\, -j}$) + Holidays$_{,\, i\, -j}$

Where $x$ is a random effect, $a$ indicates individual districts, $j$ is a lag between 1 and 30 days, and $i$ is the original date of data collection. We trained models for each time period and lag values to determine the optimal lag between $R_t$ and mobility. We assessed the different models by comparing the marginal $R^2$, which represents the contribution of fixed effects only, for different lag values and time periods. We also calculated the Median Absolute Error of each model:

$$MAE = median(|predicted_i - observed_i|)$$

For $i$ in 1...$N$ values (either predicted or observed) where $N$ is the total number of observations.

Coefficients of the Level 2 model estimate R$_t$ given inverse NPI stringency, holiday events, and the residuals of the Level 1 model (which can be interpreted as "mobility not explained by NPI stringency"). The use of inverse NPI stringency means that positive coefficients can be interpreted similarly for NPI stringency and residuals of the Level 1 model. For example, a positive coefficient indicates that R$_t$ will increase as NPI stringency decreases. Independent variables were centred and scaled for all models to allow for comparison between model coefficients.

Holiday periods included Easter, Eid al-Fitr, Eid al-Adha, and National holidays. We used the custom NPI stringency index and performed a sensitivity analysis using the OxCGRT index (Section 2 in S1 Text). We used the Vodafone mobility index in the main model since it is available in more districts, and performed a sensitivity analysis using the Google mobility index (Section 3 in S1 Text).

To determine if the association between mobility, NPIs, and Rt at different points of the epidemic is time-varying, we repeated model training for varying-length periods from 12th March to $t$ for $t$ in 19th March to 1st September (7–137 days). To understand the influence of varying time periods on model training, we also conducted a sensitivity analysis training the model in rolling fixed-length periods of 30, 60, and 90 days (Section 4 in S1 Text).

We quantified the uncertainty of model parameters using bootstrap resampling by creating 500 resampled datasets and retraining our model for each period and each resampled dataset.

We then constructed 95% and 50% bootstrap confidence intervals of model parameters by aggregating model parameter estimates for all resampled datasets trained for each period.

## Results

### COVID-19 epidemic in Ghana

Broadly, the first wave of the national COVID-19 epidemic in Ghana was characterised by an early increase in cases in March and April 2020, followed by a decline in cases over the summer and a resurgence in June and July 2020 (Fig 1). Using aggregated surveillance data for 27 districts included in the estimation of $R_t$, we observed variations in the progression of local epidemics in individual districts (Figure A in S1 Text). Patterns in each district varied, with case reports ranging from 1 to 250 cases per day, with districts reporting cases on average in 89 of 173 days. Ghana introduced a series of NPIs in response to the growing number of COVID-19 cases in March 2020 (Fig 1). On 1st April 2020, a partial lockdown was introduced in the Ashanti and Greater Accra regions requiring individuals to remain at home except for essential errands (shopping, healthcare, use of public toilets). The restrictions also prohibited inter-city travel except for essential services. Lockdown restrictions remained in place until 28th April 2020.

Both the custom and OxCGRT stringency indices reflect similar patterns in Ghana: a peak stringency coinciding with the introduction of the partial lockdown in Ashanti and Greater Accra regions, and a following reduction beginning in July 2020 (Figure B in S1 Text). Both stringency indices also identified similar dates of maximal intervention (OxCGRT: 30th March 2020, Custom: 1st April 2020).

### Changes in mobility indicators

Both Google and Vodafone mobility indicators show similar patterns in Accra and Kumasi metropolitan assemblies (the two areas for which both indicators are available), showing approximately baseline values of movement preceding the identification of the first COVID-

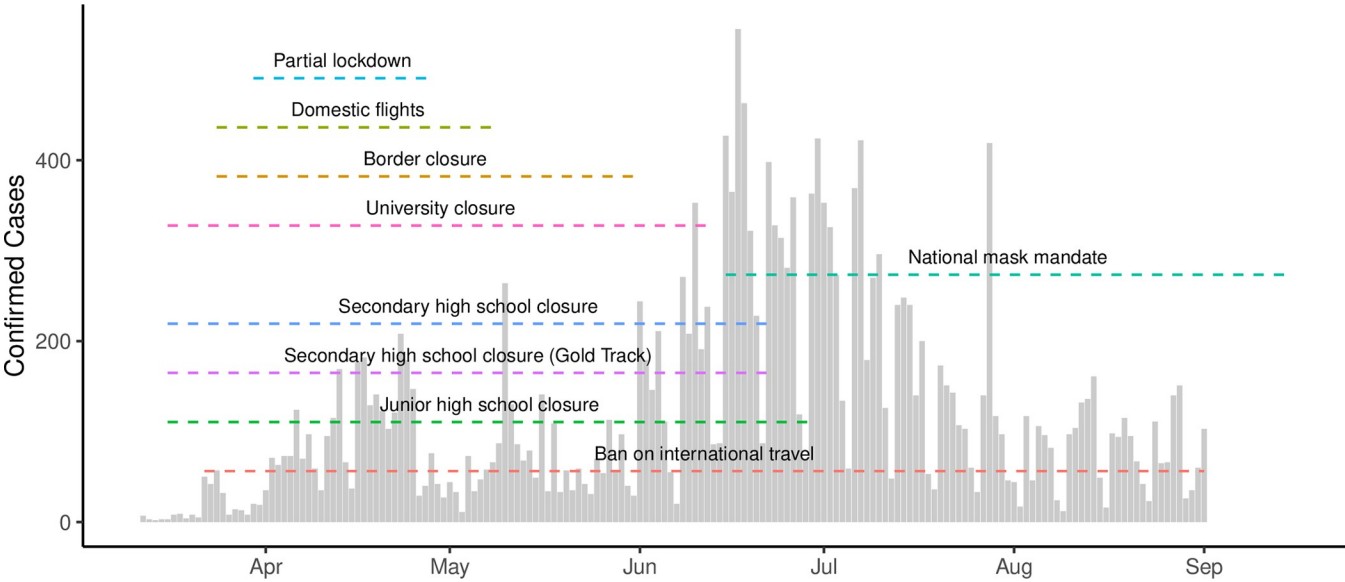

**Fig 1. Confirmed COVID-19 cases and non-pharmaceutical interventions.** The total number of confirmed cases of COVID-19 in districts included in this study. The timeline of different non-pharmaceutical interventions are indicated with dashed lines.

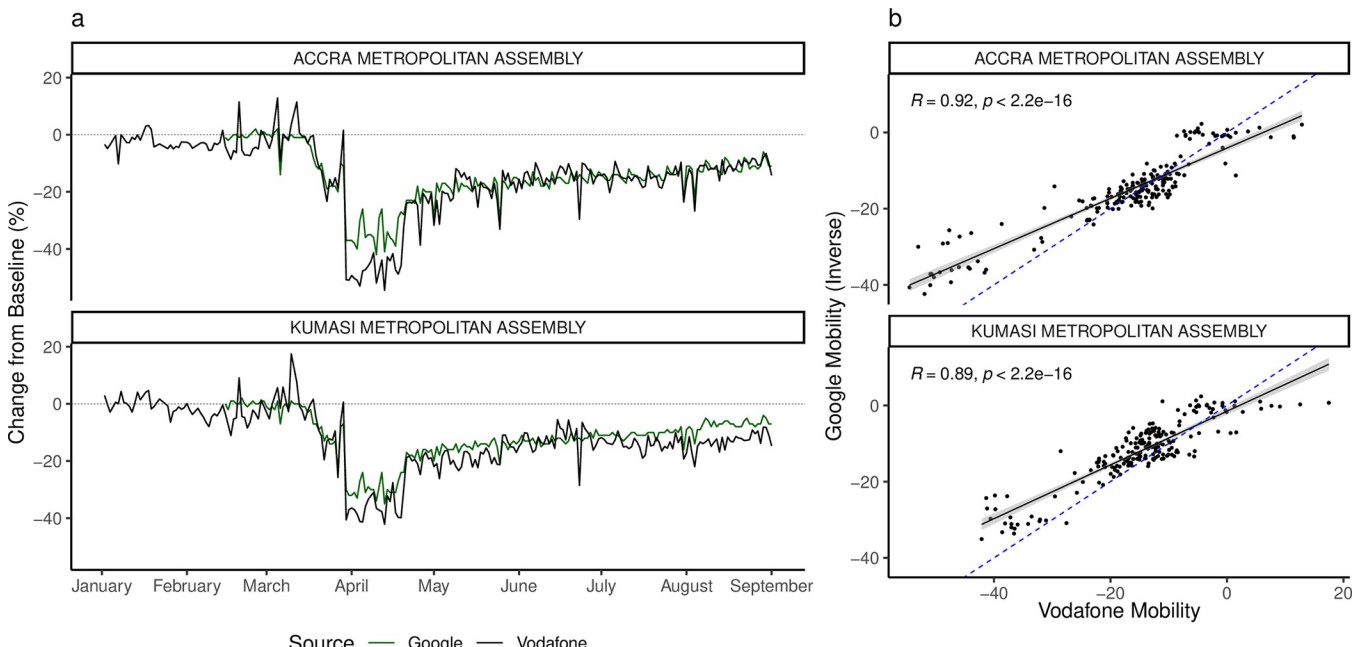

**Fig 2. Mobility indicators in Accra and Kumasi metropolitan areas.** a) A comparison of the Vodafone and Google mobility indicators in Accra and Kumasi Metropolitan Areas. b) The correlation between Vodafone and (inverse) Google mobility indicators. Blue dashed line indicates the best fit line. This shows a strong correlation between both mobility indicators across the study period. Note that these data are collected from two different sources (Google: GPS, Vodafone: CDRs) and describe different aspects of mobility (Google: activity in "residential" areas, Vodafone: travel between administrative districts).

19 cases. Both datasets show abrupt changes coinciding with the introduction of the partial lockdown, followed by a more gradual recovery (Fig 2A). Comparing the week before 30th March 2020 to the following week, mobility decreased by 24% (Vodafone) and 17% (Google) in Accra, and in Kumasi by 23% (Vodafone) and 20% (Google). We compared mobility indicators for both districts, finding strong evidence of association between Vodafone and Google mobility indicators in Accra ($R^2 = 0.92$) and Kumasi ($R^2 = 0.89$), measured between 12th March and 1st September 2020 (Fig 2B, Figure G in S1 Text).

### District-level estimates of $R_t$

After the first detection of cases on March 12th, reported cases and the number of districts reporting cases grew until the announcement of the partial lockdown (Fig 3A–3C). We found $R_t$ above 1 (indicating a growing epidemic) after the first reported COVID-19 cases in individual districts and a subsequent decline coinciding with the period of maximal interventions (Fig 3B and 3C). This was followed by an increase in $R_t$ during the summer of 2020. While district-specific epidemics followed a broad trend, transmission in individual districts was characterised by varying patterns of epidemic progression (Figure H in S1 Text). We compared $R_t$ estimates one week before and after the announcement of a partial lockdown in Ashanti and Greater Accra regions (the date of maximum intervention), finding that between 25th March and 8th April, $R_t$ decreased in 16 of the 27 districts (7 missing).

### Association between NPI stringency, mobility, and $R_t$

We found an optimal lag of mobility, NPI stringency, and holiday dates of 22 days associated with $R_t$, measured by the maximum marginal $R^2$ of the multilevel model training across all periods. Because $R_t$ estimates did not include estimated delays from infection to reporting, this

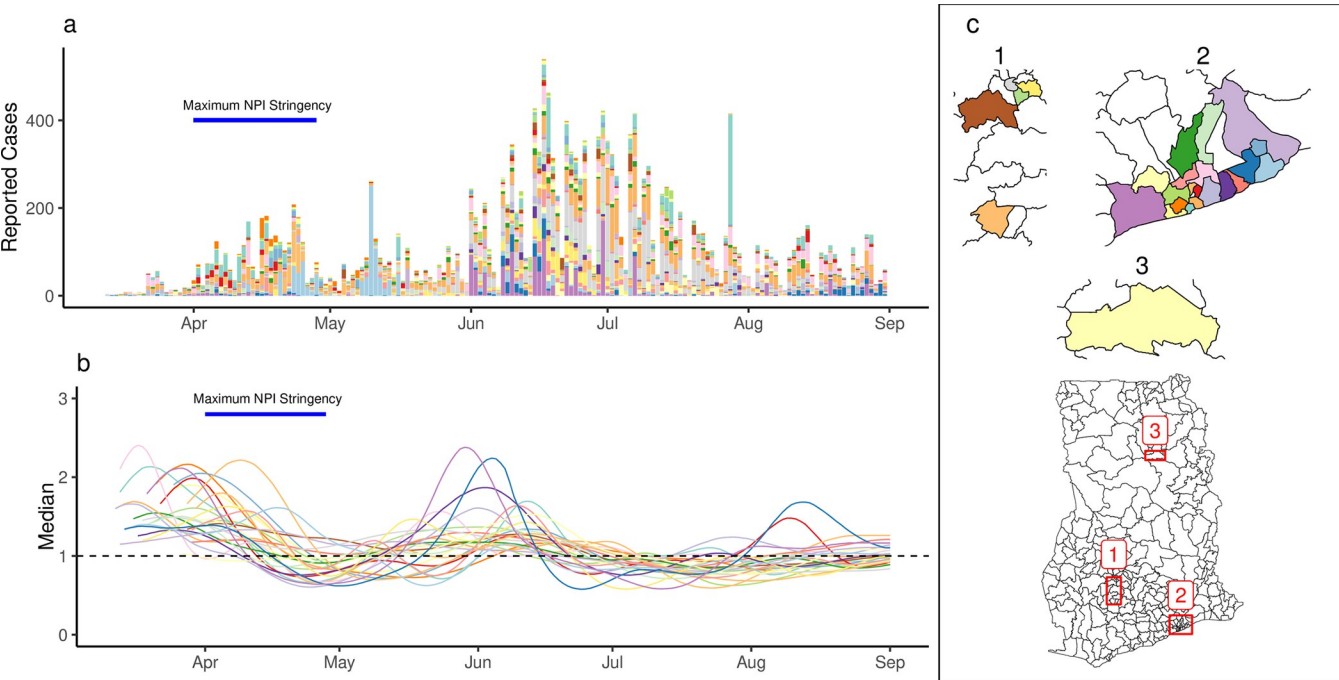

**Fig 3. Estimates of $R_t$ in individual districts.** (a) The number of reported cases in individual districts. (b) Estimates of $R_t$ for individual districts (median shown). (c) A map of districts included the analysis. Colors indicate individual districts. Basemap from the Humanitarian Data Exchange with Creative Commons Attribution for Intergovernmental Organisations Licence [35].

lag reflects the delay between mobility and NPI, infection, case detection and reporting. Sensitivity analysis using OxCGRT stringency index and Vodafone mobility indicator, as well as Google mobility indicator and Custom stringency index identified optimal lags of 21 and 19 days, respectively.

We identified correlation between mobility and inverse NPI stringency using the Level 1 model (Table B in S1 Text). We found that the Level 2 model explained a greater amount of variance in $R_t$ and that the strength of association between $R_t$ and NPI stringency was highest during the early epidemic (Fig 4A, Figure I in S1 Text). Across all training periods (173) and lag values (0–30), the maximum marginal $R^2$ was 0.51 (conditional $R^2$: 0.64) using data between March 12th and May 12th (Fig 4C). We observed higher absolute error in the beginning of case reporting in specific districts (Figure J in S1 Text). The model identified strong evidence of a positive association between $R_t$ and both NPIs and Residual Mobility (Table 1). Positive coefficients indicate an association between NPI stringency and Residual Mobility where $R_t$ increases as NPI stringency decreases and mobility increases. Note that a positive coefficient for NPI stringency results from the use of inverse stringency in the model. Throughout the study period, we observed higher uncertainty around the association between Rt and NPIs compared to the association between Rt and mobility (Fig 4B). We did not find evidence of association between $R_t$ and holidays (Table 1). For this model, we found 15 district-specific random effects distinguishable from 0 (55.6%) (Figure K in S1 Text).

The performance of the model declined through time from June to September, measured by decreasing marginal $R^2$ and increased Median Absolute Error. This reflects a period when mobility in most districts was recovering while overall, epidemics decreased. The change in model performance through time may reflect a "decoupling" of transmission from mobility and NPI stringency.

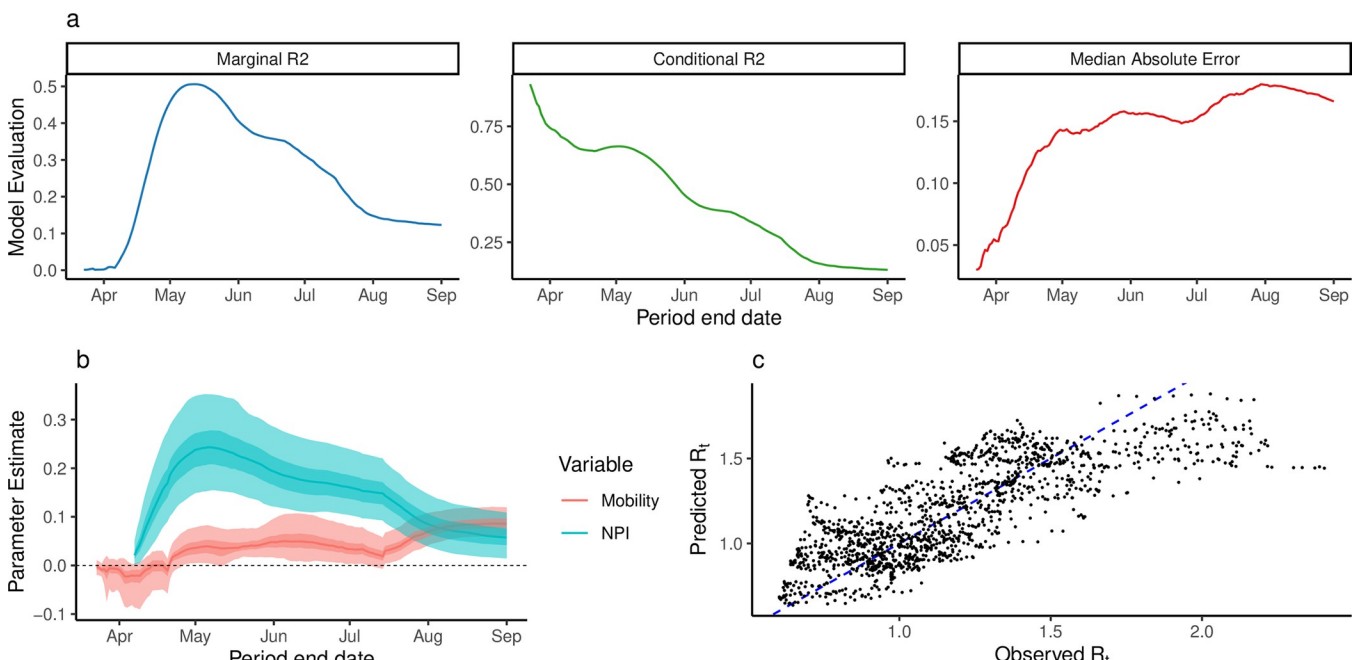

**Fig 4. Statistical analysis of $R_t$.** a) The marginal $R^2$, Conditional R2, and median absolute error of the multilevel model trained on varying-length periods through time. b) Parameter estimates of association of Rt with NPI and Mobility. Estimates shown with 50% and 95% bootstrap confidence intervals. c) Observed vs Predicted Rt for the model trained between 12th March and 12th May, 2020. Diagonal where x = y shown as blue dashed line.

Sensitivity analyses using fixed-length periods of 30, 60, and 90 days identified a similar pattern of model performance during the early epidemic and an increase in model performance in the later epidemic, relative to models trained on varying-length periods (Section 4 in S1 Text).

## Discussion

We found that $R_t$ was associated with human mobility and NPI stringency in the early stages of the COVID-19 pandemic in Ghana and that this association decreased through time. We also identified a positive association between residual mobility (mobility not explained by NPIs) and $R_t$. In our sensitivity analyses, we found similar optimal lags for both the Custom and OxCGRT stringency indices. These lags were greater than those reported in other studies, (Badr et al. for example, identified an optimal lag of 14 days between mobility and COVID-19 transmission in the USA). In sensitivity analyses included in Section 2 in S1 Text, using Vodafone and OxCGRT data, we found that the OxCGRT stringency index explained a greater amount of variance in $R_t$ in the end of the study period, which may be due to higher stringency value of the OxCGRT index in this period. We also performed sensitivity analyses using mobility indicators from two providers, Google and Vodafone Ghana, and detected similar results.

**Table 1. Regression coefficients for the multilevel model.** Regression coefficients for the multilevel model trained between 12th March and 12th May, 2020. Table shows coefficients, 95% bootstrap confidence intervals, and p values for each predictor.

| Predictors | Estimates | CI |
|---|---|---|
| NPI | 0.238 | 0.153–0.348 |
| Mobility Residuals | 0.035 | -0.001–0.075 |
| Holidays | -0.018 | -0.07–0.016 |

Between the introduction of COVID-19 in Ghana and the end of the partial lockdown, we observed a relationship between mobility indicators, NPI stringency and $R_t$ (at the maximum, our model explained approximately 64% of variation in $R_t$). The strength of this relationship decreased in June and July, and especially after August, when Ghana experienced stability in the number of reported cases and approximately constant levels of NPI stringency and Mobility. The declining relationship may indicate a disconnection between mobility and $R_t$ as the effect of mobility was mediated by other behavioural changes. It is also notable that we did not observe a decrease in mobility preceding the end of the second wave of the pandemic in July. This may indicate that mobility data is most useful in the beginning of the pandemic, when mobility patterns reflect behavioural changes relevant to disease transmission.

These findings are in line with those from high-income countries which find associations between decreases in human movement and a reduction in $R_t$ [12, 20, 36, 37] and provide evidence of the utility of mobility measures for understanding transmission in African countries. In particular, we show how human mobility and NPI stringency related to $R_t$ during the early stages of the Ghanaian COVID-19 epidemic and provide a novel analysis of subnational human mobility indicators and local disease surveillance data in a lower-middle income setting. This analysis improves our understanding of the relationship between NPIs, Mobility and the progression of COVID-19 in Ghana, and how this relationship varied through time. Future research should focus further on how human mobility indicators can be used as a proxy for social contact (and thereby transmission) and how this link changes through time. Increasing the spatial extent of case reporting data in Ghana could allow for more detailed research in districts outside of major urban areas.

We used an $R_t$ estimation method that supports uncertain generation times via a Bayesian prior with mean 3.6 days (standard deviation of mean 0.7 days) for calculating $R_t$. The use of longer generation times will lead to greater variance in estimates of $R_t$. This could translate into larger effect sizes (positive or negative) in the statistical model.

Analysing only districts with case counts which were available early in the COVID-19 epidemic in Ghana may bias our estimates towards urban populations or populations with greater disease surveillance resources. It is also not possible to determine whether the timing of the first reported cases of COVID-19 in individual districts is related to the progression of local epidemics or to the first availability of PCR testing resources in each district. The mobility indicators used in this study rely on the aggregated locations of subscribers to mobile networks (Vodafone) and users of internet services (Google). The volume and reporting of these locations may be influenced by varying patterns of mobile device usage. The demographics of users of either service may also be different from the demographic of the population of Ghana, particularly for Google data which relies on data collected from internet-connected smartphones [38]. Additionally, we used national, not district-specific indices of NPI stringency. Neither index includes intervention measures which may have been implemented in local districts but which are not recorded at a national level.

In this study, we identified evidence of positive associations between mobility, NPI stringency, and $R_t$ and show how the strength of this relationship changed through time. We found that mobility and NPI stringency was able to explain variance in $R_t$ during the early epidemic but this pattern declined as the epidemic progressed. This decline may reflect a disconnection between disease transmission and behavioural changes measured by mobility and NPI indicators. For policymakers and public health decision makers responding to the COVID-19 pandemic, our findings demonstrate that mobility and NPIs were effective for estimating disease transmission during the early epidemic, but that subsequent outbreaks may be more related to factors that are not captured in these data.

## Supporting information

**S1 Text.**
(DOCX)

## Acknowledgments

The following authors were part of the Centre for Mathematical Modelling of Infectious Disease 2019-nCoV working group. Each contributed in processing, cleaning and interpretation of data, interpreted findings, contributed to the manuscript, and approved the work for publication: Mark Jit, Rachael Pung, Thibaut Jombart, Billy J Quilty, Anna M Foss, Carl A B Pearson, Timothy W Russell, David Simons, Stefan Flasche, Graham Medley, C Julian Villabona-Arenas, Emily S Nightingale, Fabienne Krauer, Jiayao Lei, Kerry LM Wong, Jack Williams, Oliver Brady, Arminder K Deol, Yung-Wai Desmond Chan, Akira Endo, Alicia Showering, William Waites, Ciara V McCarthy, Nikos I Bosse, Kiesha Prem, Naomi R Waterlow, Yalda Jafari, Rachel Lowe, Paul Mee, Megan Auzenbergs, Kevin van Zandvoort, Joel Hellewell, Adam J Kucharski, Samuel Clifford, Mihaly Koltai, Christopher I Jarvis, James W Rudge, Fiona Yueqian Sun, W John Edmunds, Quentin J Leclerc, Simon R Procter, Matthew Quaife, Stéphane Hué, Gwenan M Knight, Nicholas G. Davies, David Hodgson, Georgia R Gore-Langton, Petra Klepac, Emilie Finch, Jon C Emery, Katherine E. Atkins, Katharine Sherratt, Alicia Rosello, Sophie R Meakin, Rein M G J Houben, James D Munday, Sebastian Funk, Lloyd A C Chapman, Frank G Sandmann, Rosanna C Barnard, Charlie Diamond, Damien C Tully, Kaja Abbas, Amy Gimma, Kathleen O'Reilly.

## Author Contributions

**Conceptualization:** Hamish Gibbs, Yang Liu, Sam Abbott, Michael Marks, Rosalind M. Eggo.

**Data curation:** Hamish Gibbs, Isaac Baffoe-Nyarko, Dennis O. Laryea, Ernest Akyereko, Patrick Kuma-Aboagye, Ivy Asantewaa Asante, Franklin Asiedu-Bekoe, Michael Marks.

**Formal analysis:** Hamish Gibbs.

**Funding acquisition:** Oriol Mitjà, William Ampofo, Michael Marks, Rosalind M. Eggo.

**Investigation:** Hamish Gibbs, Yang Liu, Rosalind M. Eggo.

**Methodology:** Hamish Gibbs, Yang Liu, Sam Abbott, Oriol Mitjà, Michael Marks, Rosalind M. Eggo.

**Project administration:** Isaac Baffoe-Nyarko, Dennis O. Laryea, Ernest Akyereko, Patrick Kuma-Aboagye, Ivy Asantewaa Asante, William Ampofo, Franklin Asiedu-Bekoe, Michael Marks, Rosalind M. Eggo.

**Resources:** Isaac Baffoe-Nyarko.

**Software:** Hamish Gibbs.

**Supervision:** Oriol Mitjà, William Ampofo, Rosalind M. Eggo.

**Validation:** Yang Liu.

**Visualization:** Hamish Gibbs.

**Writing – original draft:** Hamish Gibbs, Oriol Mitjà, Michael Marks, Rosalind M. Eggo.

**Writing – review & editing:** Yang Liu, Sam Abbott, Isaac Baffoe-Nyarko, Dennis O. Laryea, Ernest Akyereko, Patrick Kuma-Aboagye, Ivy Asantewaa Asante, William Ampofo, Franklin Asiedu-Bekoe.

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
