## [Decision Letter · Decision Letter 0]

16 Mar 2022

PGPH-D-21-01128

Association between mobility, non-pharmaceutical interventions, and COVID-19 transmission in Ghana: a modelling study using mobile phone data

Dear Dr. Gibbs,

Thank you for submitting your manuscript to PLOS Global Public Health. After careful consideration, we feel that it has merit but does not fully meet PLOS Global Public Health’s publication criteria as it currently stands. Therefore, we invite you to submit a revised version of the manuscript that addresses the points raised during the review process.

We look forward to receiving your revised manuscript.

Kind regards,

Kate Zinszer

Academic Editor

Journal Requirements:

1. Please provide separate figure files in .tif or .eps format only.  Please ensure that all files are under our size limit of 20MB.  

For more information about how to convert your figure files please see our guidelines: Once you've converted your files to .tif or .eps, please also make sure that your figures meet our format requirements

2. Please amend your detailed Financial Disclosure statement. This is published with the article, therefore should be completed in full sentences and contain the exact wording you wish to be published.

iii). State what role the funders took in the study. If the funders had no role in your study, please state: “The funders had no role in study design, data collection and analysis, decision to publish, or preparation of the manuscript.”

3. We have noticed that you have uploaded supporting information but you have not included a list of legends.  Please add a full list of legends for all supporting information files (including figures, table and data files) after the references list. 

4. Please provide us with a direct link to the base layer of the map used in Fig 3 and ensure this location is also included in the figure legend. 

Please note that, because all PLOS articles are published under a CC BY license (creativecommons.org/licenses/by/4.0/), we cannot publish proprietary maps such as Google Maps, Mapquest or other copyrighted maps. If your map was obtained from a copyrighted source please amend the figure so that the base map used is from an openly available source.

Please note that only the following CC BY licences are compatible with PLOS licence: CC BY 4.0, CC BY 2.0  and CC BY 3.0, meanwhile such licences as CC BY-ND 3.0 and others are not compatible due to additional restrictions. If you are unsure whether you can use a map or not, please do reach out and we will be able to help you. 

The following websites are good examples of where you can source open access or public domain maps:

Additional Editor Comments (if provided):

The authors present a very interesting analysis with novel data sources. There are comments that need to be addressed before it is suitable for publication. The last paragraph in the introduction is more suitable for the methods section, the objective needs to be more clearly specified, there needs to be some information on the SARS-CoV-2 testing strategy and accessibility during the study period in the methods, and clarity on how the model performance was evaluated.

Reviewers' comments:

Reviewer's Responses to Questions

**Comments to the Author**

1. Does this manuscript meet PLOS Global Public Health’s publication criteria? Is the manuscript technically sound, and do the data support the conclusions? The manuscript must describe methodologically and ethically rigorous research with conclusions that are appropriately drawn based on the data presented.

Reviewer #1: Yes

Reviewer #2: Yes

2. Has the statistical analysis been performed appropriately and rigorously?

Reviewer #1: Yes

Reviewer #2: Yes

3. Have the authors made all data underlying the findings in their manuscript fully available (please refer to the Data Availability Statement at the start of the manuscript PDF file)?

Reviewer #1: Yes

Reviewer #2: Yes

4. Is the manuscript presented in an intelligible fashion and written in standard English?

Reviewer #1: Yes

Reviewer #2: Yes

5. Review Comments to the Author

Reviewer #1: I suggest making clear the objective: In this article, the objective is to assess the combination of human mobility and NPI data to estimate their relationship over time with the progress of the COVID-19 97 epidemic in Ghana.

The analysis was carefully carried out, considering adjustments for holidays

Despite not including the period with Alpha or Beta variants, the manuscript had significant results for the start of the pandemic. Mobility and NPI stringency was able to explain variance in Rt during the early epidemic but this pattern declined as the epidemic progressed.

I suggest reviewing the formatting of table 1. It's like a picture.

Reviewer #2: This manuscript studies the relationship between human mobility, non-pharmaceutical interventions, and COVID transmission in Ghana. This work shows well-conducted research and the manuscript is well-written. I believe some points on the validation of the experimental results and the uncertainty characterization of the model should be clarified before it could be published.

First, it is unclear to me what kind of validation was used in the experiments to show the association between the case increases/decreases and the non-pharmaceutical interventions. Are these results temporal cross-validated using future datasets outside of the training set?

Second, how is the uncertainty in the proposed method evaluated and controlled in this paper? For example, Shea et al (2020) (Shea, Katriona, et al. "COVID-19 reopening strategies at the county level in the face of uncertainty: Multiple Models for Outbreak Decision Support." medRxiv (2020).) aggregated multiple models to capture the uncertainty of individual models in characterizing the association between the outbreak of COVID and interventions by policy makers.

6. PLOS authors have the option to publish the peer review history of their article (what does this mean?). If published, this will include your full peer review and any attached files.

**Do you want your identity to be public for this peer review?** For information about this choice, including consent withdrawal, please see our Privacy Policy.

Reviewer #1: No

Reviewer #2: No

---

## [Decision Letter · Decision Letter 1]

26 Jul 2022

Association between mobility, non-pharmaceutical interventions, and COVID-19 transmission in Ghana: a modelling study using mobile phone data

PGPH-D-21-01128R1

Dear Mr. Gibbs,

We are pleased to inform you that your manuscript 'Association between mobility, non-pharmaceutical interventions, and COVID-19 transmission in Ghana: a modelling study using mobile phone data' has been provisionally accepted for publication in PLOS Global Public Health.

Best regards,

Kate Zinszer

Academic Editor

Reviewer Comments (if any, and for reference):

Reviewer's Responses to Questions

**Comments to the Author**

1. If the authors have adequately addressed your comments raised in a previous round of review and you feel that this manuscript is now acceptable for publication, you may indicate that here to bypass the “Comments to the Author” section, enter your conflict of interest statement in the “Confidential to Editor” section, and submit your "Accept" recommendation.

Reviewer #1: All comments have been addressed

Reviewer #2: All comments have been addressed

2. Does this manuscript meet PLOS Global Public Health’s publication criteria? Is the manuscript technically sound, and do the data support the conclusions? The manuscript must describe methodologically and ethically rigorous research with conclusions that are appropriately drawn based on the data presented.

Reviewer #1: Yes

Reviewer #2: Yes

3. Has the statistical analysis been performed appropriately and rigorously?

Reviewer #1: Yes

Reviewer #2: Yes

4. Have the authors made all data underlying the findings in their manuscript fully available (please refer to the Data Availability Statement at the start of the manuscript PDF file)?

Reviewer #1: Yes

Reviewer #2: (No Response)

5. Is the manuscript presented in an intelligible fashion and written in standard English?

Reviewer #1: Yes

Reviewer #2: Yes

6. Review Comments to the Author

Reviewer #1: Relevant manuscript. The authors will make the modifications. I suggest accept.

Reviewer #2: I appreciate the authors for addressing my questions in their revision.

7. PLOS authors have the option to publish the peer review history of their article (what does this mean?). If published, this will include your full peer review and any attached files.

**Do you want your identity to be public for this peer review?** For information about this choice, including consent withdrawal, please see our Privacy Policy.

Reviewer #1: **Yes: **Luana Vieira Toledo

Reviewer #2: No
